# Diel transcriptional responses of coral-Symbiodiniaceae holobiont to elevated temperature
Sanqiang Gong[1,2], Jiayuan Liang[2], Lijia Xu[3], Yongzhi Wang[3], Jun Li[1], Xuejie Jin[1], Kefu Yu ®[2,4] ✉ & Yuehuan Zhang[1] ✉

Coral exhibits diel rhythms in behavior and gene transcription. However, the influence of elevated temperature, a key factor causing coral bleaching, on these rhythms remains poorly understood. To address this, we examined physiological, metabolic, and gene transcription oscillations in the *Acropora tenuis-Cladocopium* sp. holobiont under constant darkness (DD), light-dark cycle (LD), and LD with elevated temperature (HLD). Under LD, the values of photosystem II efficiency, reactive oxygen species leakage, and lipid peroxidation exhibited significant diel oscillations. These oscillations were further amplified during coral bleaching under HLD. Gene transcription analysis identified 24-hour rhythms for specific genes in both coral and Symbiodiniaceae under LD. Notably, these rhythms were disrupted in coral and shifted in Symbiodiniaceae under HLD. Importantly, we identified over 20 clock or clock-controlled genes in this holobiont. Specifically, we suggested *CIPC* (CLOCK-interacting pacemaker-like) gene as a core clock gene in coral. We observed that the transcription of two abundant rhythmic genes encoding glycoside hydrolases (*CBM21*) and heme-binding protein (*SOUL*) were dysregulated by elevated temperature. These findings indicate that elevated temperatures disrupt diel gene transcription rhythms in the coral-Symbiodiniaceae holobiont, affecting essential symbiosis processes, such as carbohydrate utilization and redox homeostasis. These disruptions may contribute to the thermal bleaching of coral.

The solar, tidal, and lunar cycles have significant effects on the behavior, metabolism and gene transcription of organisms in marine ecosystems[1]. These diel rhythms lead to the development of circadian clocks, which help regulate various biological processes and improve the fitness and survival of different kingdoms of life, including cyanobacteria[2], fungi[3], plant[4] and animal[5,6].

Coral reefs, intricate ecosystems providing habitants for abundant species, are primarily constructed by reef-building corals[7]. The success of coral reefs in nutrient-poor seawater is attributed to the mutually beneficial symbiotic relationship between coral and photosynthetic Symbiodiniaceae[8]. This partnership offers an avenue to explore the influence of diel rhythm on the coordination of the symbiosis between coral and its hosted Symbiodiniaceae[9]. Some studies have observed rhythmic and coordinated behavior between coral and Symbiodiniaceae[10–13]. For example, one research

on sea anemone *Aiptasia* showed that symbiotic Symbiodiniaceae can influence the transcriptional rhythm of host[14]. Similarly, study on the mesophotic coral *Euphyllia paradivisa* has shown that this species has an internal clock, which is influenced by its symbiotic Symbiodiniaceae[15]. Rhythmic processes were also observed in symbiotic Symbiodiniaceae, proposed the presence of endogenous clocks in both coral and its hosted Symbiodiniaceae[10–13].

It is well known that elevated temperatures significantly impact coral and Symbiodiniaceae growth and survival[16,17]. Coral bleaching, a major threat to reef ecosystems, is primarily driven by elevated temperature[18]. It is well established that elevated temperature has a great effect on the diel rhythms of gene transcription in plants (such as soybean)[19], animals (such as *Drosophila*)[20], and bacteria (such as *Klebsiella aerogenes*)[21,22]. However, the influence of elevated temperatures on the rhythms of gene

[1]Key Laboratory of Tropical Marine Bio-resources and Ecology & Guangdong Provincial Key Laboratory of Applied Marine Biology, South China Sea Institute of Oceanology, Chinese Academy of Sciences, Guangzhou, 510301, China. [2]Guangxi Laboratory on the Study of Coral Reefs in the South China Sea, Coral Reef Research Center of China, School of Marine Sciences, Guangxi University, Nanning, 530004, China. [3]South China Institute of Environmental Sciences, The Ministry of Ecology and Environment of PRC, Guangzhou, 510530, China. [4]Southern Marine Science and Engineering Guangdong Laboratory (Guangzhou), Guangzhou, 511458, China. ✉e-mail: kefuyu@scsio.ac.cn; yhzhang@scsio.ac.cn

**Fig. 1 | Physiology and metabolism rhythms of *A. tenuis-Cladocopium* sp. holobiont.** The experimental design involved cultivating reef-building *A. tenuis* samples under three light conditions: DD, LD and HLD (**a**). The horizontal boxes represent the light and dark phases during sampling, with blue boxes indicating light and gray boxes indicating dark. The arrows (pointing upwards) mark the sampling time points. Sampling commenced at midnight (time 0 = 24:00) on November 13, 2021, and continued at 6-h intervals for 48–72 h. The oscillation plots depict variations in *Symbiodiniaceae* cell densities (**b**), photochemical efficiency of Photosystem II (*Fv/Fm*) (**c**), relative release of reactive oxygen species (ROS) by symbiotic *Symbiodiniaceae* (**d**), host lipid peroxidation levels (**e**). Error bars represent mean values ± standard deviation (SD), with *n* = 9.

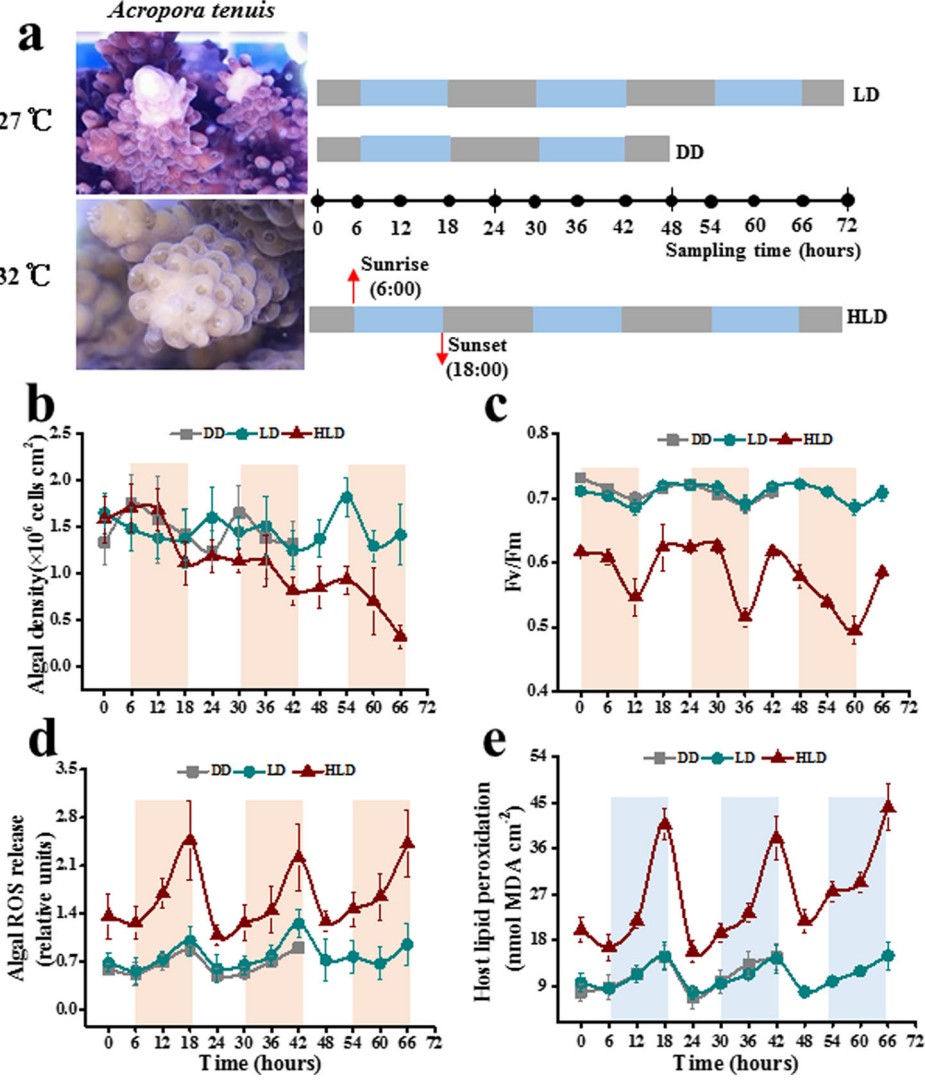

transcription that possibley regulate symbiosis during coral bleaching is poorly understood.

To address this gap, our study examines changes in physiology, biochemistry, cellular processes, and transcriptome over a 48–72 h period under three conditions: constant darkness (DD), light/dark cycle (LD), and LD with elevated temperature (HLD). Using *Acropora tenuis*-Symbiodiniaceae (*Cladocopium* sp. genus) as a model holobiont, we aims to investigate how elevated temperatures affect the diel rhythms of coral holobiont. The availability of sequenced genomes for both *A. tenuis* and *Cladocopium* makes this holobiont an ideal model for studying symbiosis[23,24]. Through rhythmic analysis, we identified the 24-h periods of gene transcription patterns in both *A. tenuis* and its symbiotic Symbiodiniaceae, revealing distinct diel gene transcription patterns under LD and HLD conditions.

## Results
### Physiology and metabolism rhythms of coral-Symbiodiniaceae holobiont
Colonies of *A. tenuis* were subjected to three conditions: constant darkness at 27 °C (DD), light/dark cycle at 27 °C (LD), and LD at 32 °C (HLD) (Fig. 1a). Visual observations indicated thermal bleaching in *A. tenuis* colonies (Fig. 1a), with an 86% reduction in algal symbiont density after 72 h of HLD cultivation (Fig. 1b). The thermal bleaching of *A. tenuis* was further linked to significant diel rhythmic oscillations (24-h) of photosynthetic efficiency (*Fv/Fm*, Fig. 1c), reactive oxygen species (ROS, Fig. 1d) leakage, and lipid peroxidation (Fig. 1e) in the *A. tenuis* holobiont (Cosinor

algorithm, *p*-values < 0.05, Table 1, Supplementary Data 1). ROS leakage and host lipid peroxidation peaked at dusk (18:00), while *Fv/Fm* peaked at midday (12:00). Elevated temperature amplified oscillation amplitude of ROS leakage, lipid peroxidation, and *Fv/Fm*. Compared to LD, HLD resulted in a decrease in *Fv/Fm* (7–30%) and increases in ROS (>43%) and lipid peroxidation (48–63%).

### Transcriptome rhythmic analysis of coral-Symbiodiniaceae holobiont
To reveal the molecular causes behind the observed phenotype, physiology and metabolism, we further examined the rhythmic genes and their oscillation patterns in the *A. tenuis*-Symbiodiniaceae holobiont at 6-h intervals under DD, LD and HLD conditions.

Firstly, a rigorous analysis was conducted to identify rhythmic genes (JTK + Cosinor, *q*-values < 0.05), resulting in 0, 34 and 0 genes identified in the coral host under DD, LD, and HLD conditions, respectively (Supplementary Data 2). For the symbiont, we found 0, 1 and 0 rhythmic genes under the same conditions (Supplementary Data 3). Considering the difficulty to identify rhythmic genes with *q*-values < 0.05, we then opted for *p*-values < 0.02 to identify rhythmic genes, acknowledging the risk for false-positive results. Under DD, LD, and HLD conditions, 100, 522, and 72 rhythmic genes were identified in the host, respectively (JTK + Cosinor, *p*-values < 0.02, Fig. 2a, Supplementary Data 2). In contrast, 75, 64, and 93 rhythmic genes were detected in the symbiont (JTK + Cosinor, *p*-values < 0.02, Fig. 2c, Supplementary Data 3). Among the identified rhythmic genes,

25 exhibited oscillatory transcription in the host under both DD and LD conditions. In the symbiont, we also detected several rhythmic genes, which exhibiting similar oscillations under both DD and LD conditions. These genes were designated as candidate clock or clock-controlled genes (Supplementary Data 4).

Analysis of the identified rhythmic genes from DD, LD and HLD conditions (JTK + Cosinor, *p*-values < 0.02, after deduplication) and their heatmaps (Fig. 2b, d) revealed that the coral host displayed more robust diel gene transcription than its symbiont under LD. However, HLD resulted in a loss of rhythmicity in the majority of host genes.

Furthermore, we observed that the overall oscillation patterns of rhythmic genes identified under LD (JTK + Cosinor, *p*-values < 0.02) were disrupted in the coral (Fig. 3a) and shifted in the symbiont under HLD (Fig. 3b). These disordered oscillation patterns were further supported by Gaussian mixture model (GMM) (Fig. 3c, d) clustering of rhythmic genes

**Table 1 | Rhythmicity test of algal density, photosynthetic efficiency, reactive oxygen species (ROS) leakage, and lipid peroxidation in the *A. tenuis* holobiont (Cosinor algorithm) under DD, LD and HLD conditions**

|  | *p*-value (Cosinor) | | |
| --- | --- | --- | --- |
|  | **DD** | **LD** | **HLD** |
| Algal dencity | 0.072 | 0.110 | 0.345 |
| Fv/Fm | 0.031 | 0.014 | 0.039 |
| ROS | 0.005 | 0.014 | 0.004 |
| Lipid peroxidation | 0.012 | 0.005 | 0.023 |

identified under LD and HLD (JTK + Cosinor, *p*-values < 0.02), respectively. It was observed that most rhythmic genes (>90%) had high transcriptional abundance (peak) at dawn (6:00) or midday (12:00) in both the coral host and symbiont under LD. Conversely, under HLD, the patterns were altered, with 84% of rhythmic genes in the symbiont peaking at dusk (18:00).

## The oscillation of candidate clock or clock-controlled genes in coral holobiont under elevated temperature

In the coral host, rhythmic genes essential to the circadian clock (possibly clock genes-C) were identified (Fig. 4, Supplementary Data 4), including *CIPC* (CLOCK-interacting pacemaker-like), *Cry1* (cryptochrome 1), *phrB* (Deoxyribodipyrimidine photolyase). The *CLOCK*(Circadian locomoter output cycles kaput protein) and *BMAL1*(Aryl hydrocarbon receptor nuclear translocator) genes were actively transcribed but did not exhibit significant rhythmicity. *Cry1* peaked at midday, while *CIPC* and *phrB* peaked at dawn.

Additional rhythmic genes (possibly clock-controlled genes-CCG) (Fig. 5, Supplementary Fig. S1, Supplementary Data 4) included *ANP1* (Atrial natriuretic peptide receptor 1), *Ang1* (Angiopoietin-1 receptor), *DHRS12* (Dehydrogenases with different specificities) *TEF* (Thyrotroph embryonic factor-like), *HEY* (Hairy and enhancer of split), *UTP18* (U3 small nucleolar RNA-associated protein 18), *CBM21* (Carbohydrate/starch-binding module (family 21), *PNPO* (Pyridoxamine-phosphate oxidase), *SOUL* (SOUL heme-binding protein), *LX3* (Legumain-like isoform X3) and and others, exhibiting significant rhythmic oscillations under both DD and LD conditions. These genes involved in diverse processes such as signaling, transcription regulation, translation, cofactor and lipid metabolism, and transportation (Fig. 5).

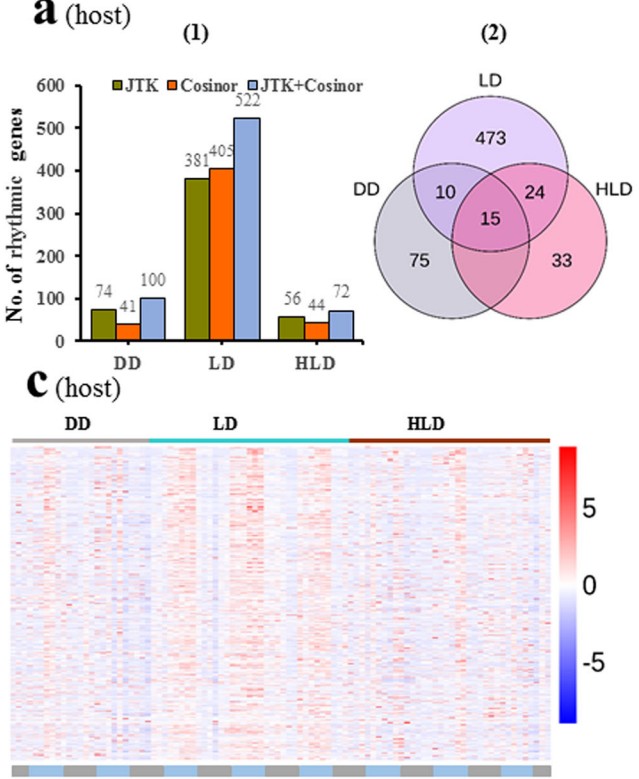

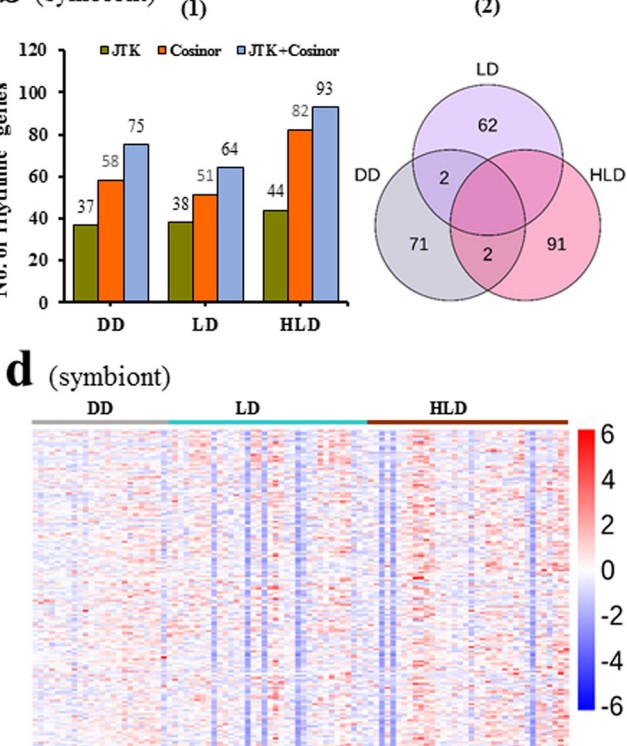

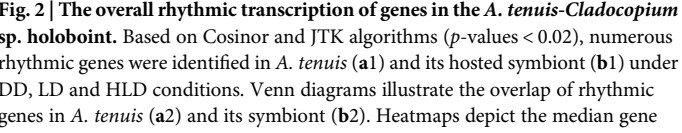

**Fig. 2 | The overall rhythmic transcription of genes in the *A. tenuis-Cladocopium* sp. holoboint.** Based on Cosinor and JTK algorithms (*p*-values < 0.02), numerous rhythmic genes were identified in *A. tenuis* (**a**1) and its hosted symbiont (**b**1) under DD, LD and HLD conditions. Venn diagrams illustrate the overlap of rhythmic genes in *A. tenuis* (**a**2) and its symbiont (**b**2). Heatmaps depict the median gene expression values of rhythmic genes identified from DD, LD and HLD conditions (JTK + Cosinor, *p*-values < 0.02, after deduplication) over time points (*n* = 3 biological replicates), where each row represents a rhythmic gene in the host (**c**) or symbiont (**d**).

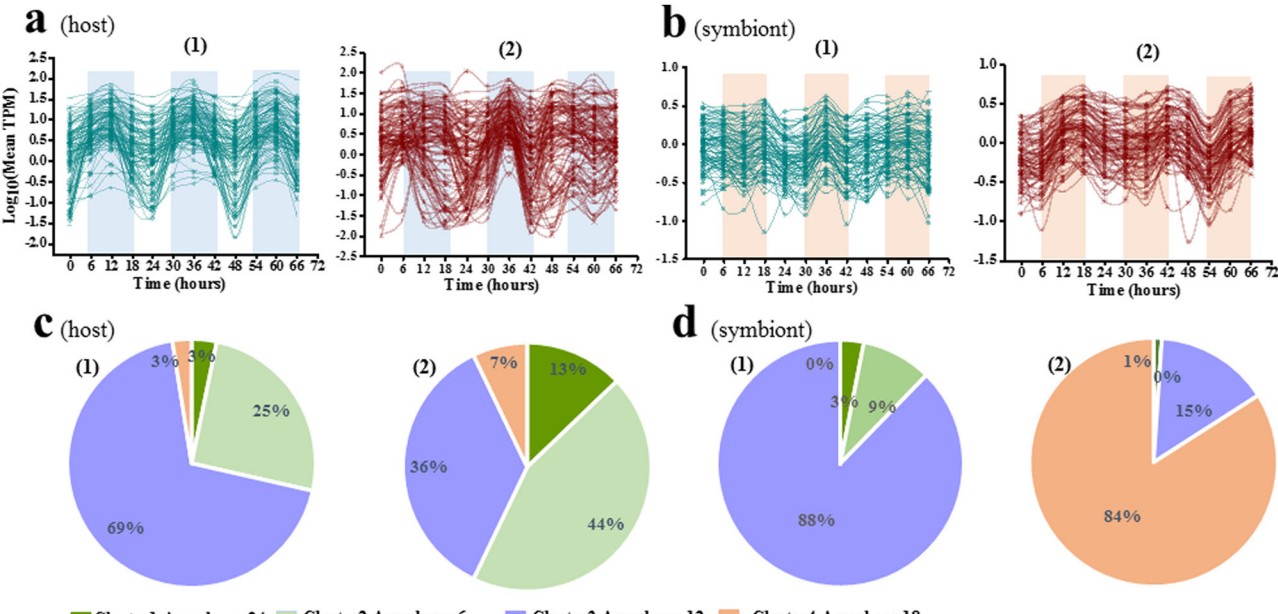

**Fig. 3 | Overall oscillation of rhythmic genes in the *A. tenuis-Cladocopium* sp. holobiont.** The overall oscillation of rhythmic genes identified under LD (JTK + Cosinor, *p*-values < 0.02) in the host (a1-LD, a2-HLD) and symbiont (b1-LD, b2-HLD) under different conditions. Lines are based on sinusoidal function. the lines are fitted using a sinusoidal function. Pie charts present the distribution (percentages) of rhythmic genes identified under LD and HLD conditions (JTK + Cosinor, *p*-values < 0.02) within different clusters (determined by Gaussian mixture model) for both the host (**c**) or symbiont (**d**).

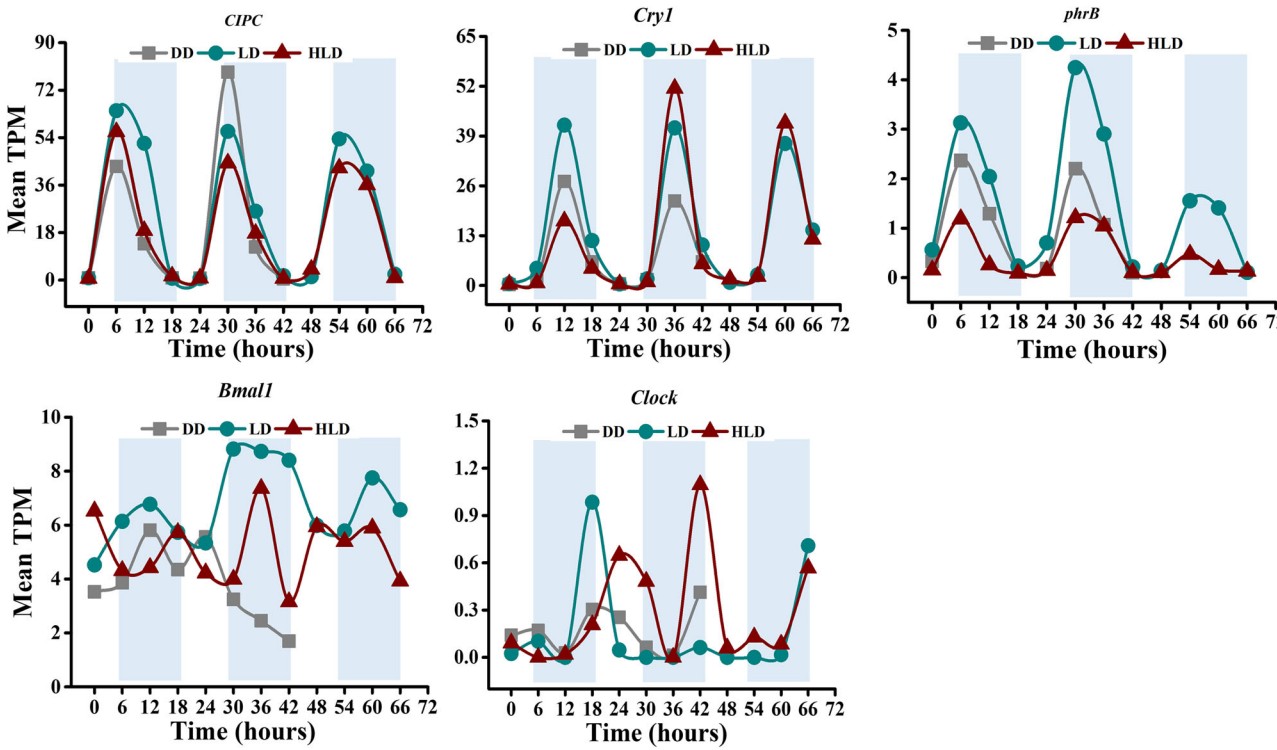

**Fig. 4 | Diel oscillation of candidate clock genes in the *A. tenuis-Cladocopium* sp. holoboint.** Oscillation plots for candidate circadian clock genes cycling with a 24-h period in host under DD, LD and HLD conditions. The x-axis of the plots represents the experimental time course (with time 0 set at 24:00), while the y-axis displays normalized expression levels. Note that the scales differ across plots. Data points are presented as mean values (*n* = 3 biological replicates).

Notably, the *CBM21* and *SOUL* genes showed the highest transcription abundance, implicated in polysaccharide utilization and cellular oxidation-reduction equilibrium, respectively. Intriguingly, these genes exhibited comparable rhythmic patterns in LD and HLD conditions (Fig. 5, Supplementary Fig. S1). However, elevated temperature reduced the transcription abundance of most core rhythmic genes at their peaks.

In the symbiont, several rhythmic genes (possibly clock-controlled genes-CCG) were observed, including *APA1* (Aspartic proteinase A1), *BMY1* (Beta-amylase), *P4H* (Prolyl 4-hydroxylase), *ARSB* (N-acetylgalactosamine 4-sulfatase), *SLC* (Solute carrier family) (Fig. 5, Supplementary Fig. S1, Supplementary Data 4). These genes may be involved in carbohydrate metabolism and transport (Fig. 5). Notably, the *BMY1* gene,

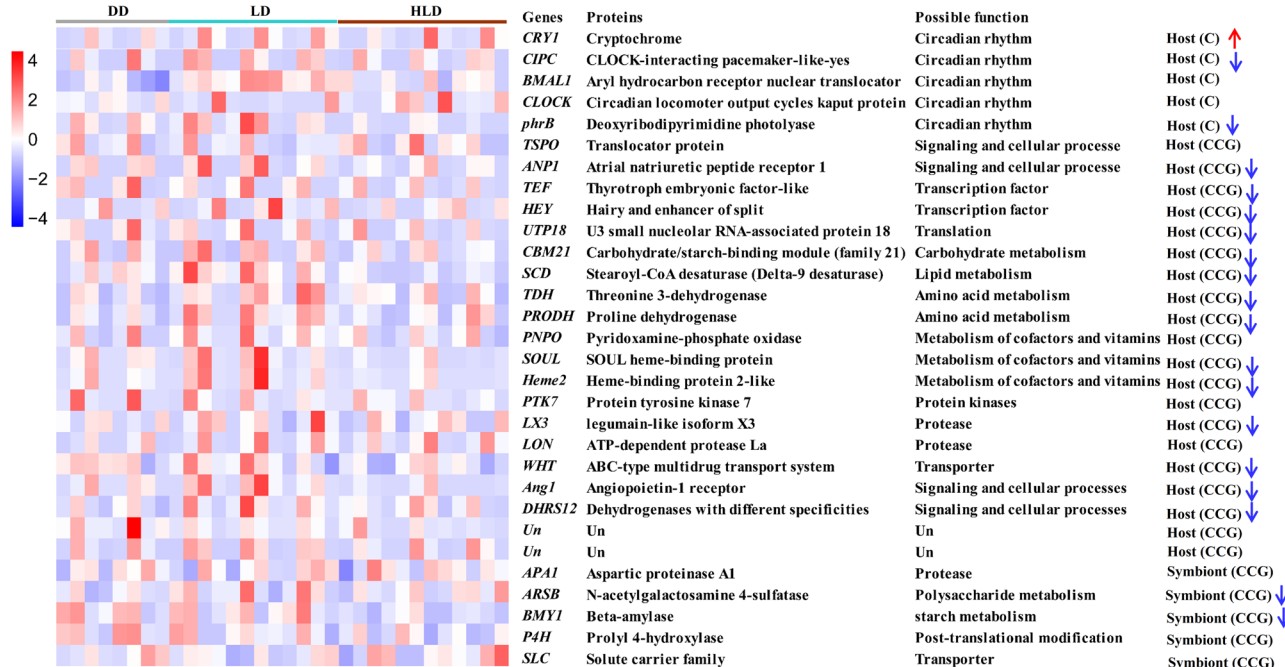

**Fig. 5 | The gene function of candidate clock or clock-controlled genes in the *A. tenuis-Cladocopium* sp. holobiont.** Heatmap visualizes the median gene expression values across time points (*n* = 3 biological replicates). Each row represents a rhythmic gene in the host or symbiont under DD, LD and HLD conditions. A list of genes and their potential functions is provided to the right of the heatmap. Up and down arrows represent increased and decreased gene transcription levels under elevated temperature. The capital letters of C and CCG represent candidate clock gene, and clock-controlled gene, respectively.

involved in polysaccharide metabolism, showed significant rhythmic oscillations under all conditions. However, its transcription abundance at peaks was diminished by elevated temperature.

## Discussion

### Oscillation of candidate clock genes in coral-Symbiodiniaceae holobiont under elevated temperature

In animal models, a CLOCK-BMAL1 heterodimer activates the transcription of its negative regulators, such as *Cry*, *Per*, *Tim* genes[25,26]. This study identifies active transcription of clock genes (*CLOCK*, *BMAL1*, *Cry*) in *A. tenuis* as well. Notably, this study reports the first identification of the *CIPC* gene in *A. tenuis*. *CIPC* encodes the CLOCK-interacting pacemaker-like protein, a newly discovered protein that functions as a circadian feedback loop negative regulator in mammals[27]. In *A. tenuis*, *CIPC* exhibits the highest transcriptional abundance among clock genes and displays a robust rhythmic oscillation, peaking at dawn under both DD and LD conditions, supporting its role as a core clock gene in coral.

The *Cry* and *phrB* genes encode evolutionarily related flavoproteins, cryptochrome and deoxyribodipyrimidine photolyase, respectively. Cryptochromes regulate growth, development, and the circadian clock in plants and animals[28,29], while photolyases repair UV-induced DNA damage in a light-dependent manner[30]. In the *A. tenuis*, both *Cry* (*Cry1*) and *phrB* genes show rhythmic oscillations under DD and LD conditions. *Cry* (*Cry1*) peaks at midday, while *phrB* peaks at dawn. While diel rhythmic oscillations of *Cry* genes have been reported previously in corals[9,13,14], this study is the first to report the diel rhythmic oscillation of the *phrB* gene. Interestingly, *CIPC*, *Cry1*, and *phrB* genes exhibit similar rhythmic oscillation patterns in both LD and HLD conditions. However, their peak transcription levels were changed by elevated temperature, suggesting that coral host circadian clock gene transcription is buffered against temperature changes. Our results indicated that the negative feedback in the coral circadian clock is probably divided into distinct pathways, and that the addition of the new clock genes, such as *CIPC and phrB*, has contributed to the complexity of coral clocks.

In photosynthetic organisms, including algae, the circadian clock comprises a large number of transcription factors organized in multiple feedback loops, including clock genes of *MYB* (MYB-related transcription factors), *LHY* (LATE ELONGATED HYPOCOTYL), *CCA1* (CIRCADIAN CLOCK ASSOCIATED 1), *TOC1* (TIMING OF CAB2 EXPRESSION1) families[4,31,32]. However, we does not detect rhythmic genes in coral-hosted Symbiodiniaceae that have been established as clock genes in photosynthetic organisms. This could be due to weak diel rhythmicity in symbiotic Symbiodiniaceae or host-mediated effects on the diel rhythmicity of symbiotic Symbiodiniaceae, requiring further investigation.

### Diel rhythmic oscillation of genes related to coral bleaching

Coral bleaching is primarily driven by elevated temperature, but the behind mechanisms remain elusive due to the complexity of the coral-Symbiodiniaceae holobiont[33]. In this study, significant rhythmic oscillations in photosynthetic efficiency (*Fv/Fm*), reactive oxygen species (ROS) leakage, and lipid peroxidation were observed in the *A. tenuis*, particularly during coral bleaching under elevated temperature. Previous studies has highlighted the importance of diel rhythmicity in coral bleaching. Baird et al. demonstrated that daytime bleaching is more pronounced than nighttime bleaching, supporting the role of diel cycles in coral bleaching responses[34]. Wooldridge and Levy observed that antioxidant defenses in corals exhibit diel rhythms, which may influence their resilience to oxidative damage during bleaching[35,36]. Our findings corroborate with their observations and provide further evidence for the role of diel rhythmicity in photosynthetic inhibition, oxidative damage of coral holobiont. We observed that the *Fv/Fm* exhibited its lowest oscillation value at midday, while ROS leakage from isolated algal symbionts and lipid peroxidation levels in the host peaked at dusk under elevated temperature. This suggests that increased oxidative damage to the coral host resulting from ROS leakage lags behind photosynthetic inhibition in symbiont.

Gene transcription analysis under LD cycles revealed diel rhythms in specific genes of both coral and Symbiodiniaceae. However, we noted that elevated temperature disrupted this diel rhythmicity of gene transcription

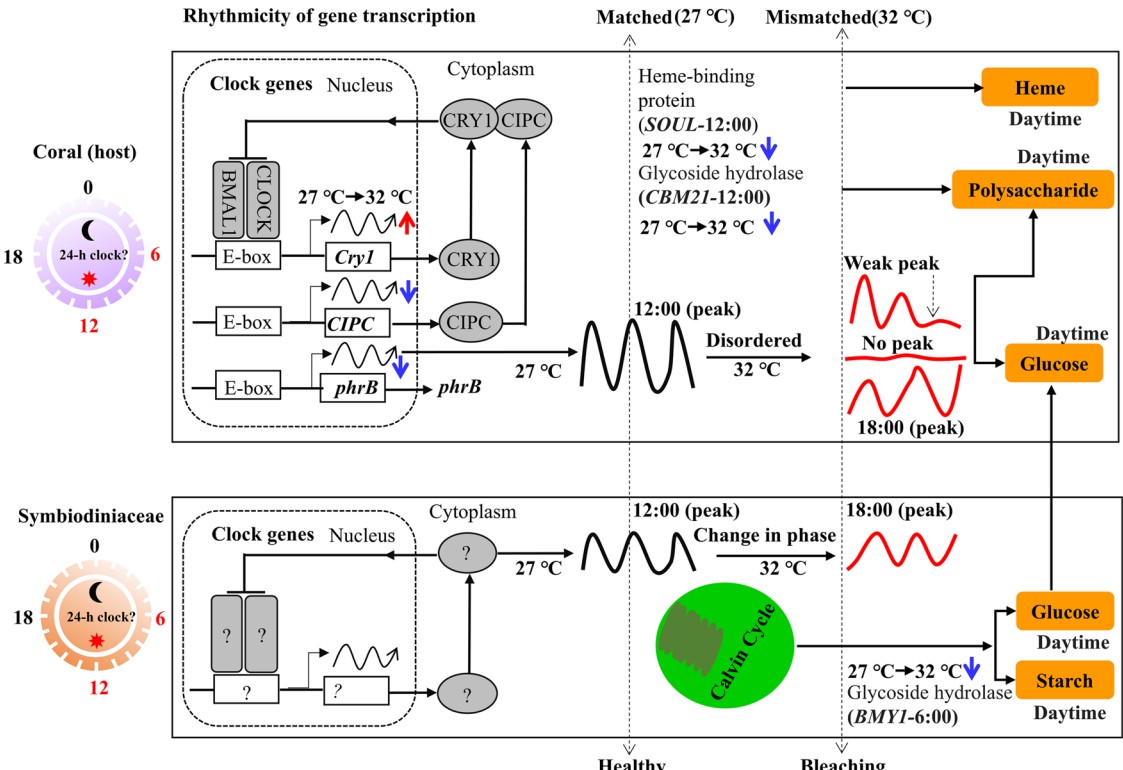

**Fig. 6 | Conceptual summary of rhythmic gene transcription in *A. tenuis* and its hosted *Cladocopium* sp. under elevated temperature, causing coral bleaching.** It illustrate distinct response of *A. tenuis* and its hosted *Cladocopium* sp. to elevated temperature in rhythmic patterns. The response of candidate circadian clock genes and several circadian clock-controlled genes (involved polysaccharide utilization and cellular redox equilibrium, such as *CLOCK, BMAL1, CIPC, Cry1, phrB CMB21, BYM1* and *SOUL* was illustrated.

observed under LD in both coral and Symbiodiniaceae. Notably, the transcription abundance of several clock-controlled genes (e.g., *ANP1, Ang1, DHRS12, TEF, HEY, UTP18, CBM21, PNPO, SOUL, LX3 BMY1, P4H, ARSB* and *SLC*) involved in signaling, cellular processes, transcription regulation, translation, cofactor metabolism, lipid metabolism, carbohydrate metabolism, transporter processes, and stress responses was altered by elevated temperature. The results align with the findings in model organisms[9,19,37], which have consistently demonstrated that elevated temperature disrupts the diel rhythms of gene transcription in coral and Symbiodiniaceae. Moreover, the alteration of clock-controlled genes involved in various cellular processes is a common observation across these studies, suggesting a critical role for temperature in regulating circadian rhythmicity and overall physiological responses in these organisms[20,38]. Among the identified core rhythmic genes in *A. tenuis*, *CBM21* and *SOUL* participating in polysaccharide utilization and cellular redox equilibrium[39–41], exhibited the highest transcriptional abundance. Their reduced transcriptional abundance under elevated temperature implies that diel rhythmicity in polysaccharide utilization and cellular redox equilibrium may be negatively affected in the coral host. In the symbiont, the *BMY1* gene (a possible circadian clock controlled gene), associated with polysaccharide metabolism[42], also showed reduced transcriptional abundance under elevated temperature, indicating that diel rhythmic polysaccharide utilization in symbiont may be impaired as well.

In all, this study provide further evidence for the role of diel rhythmicity in coral bleaching (Fig. 6). The present findings suggest that the coral host and its symbiont maintain a synchronized rhythmicity in gene transcription under normal conditions, which supports symbiosis. However, elevated temperatures disrupt this rhythmicity, potentially contributing to thermal bleaching of coral. Importantly, the identification of rhythmic genes and their oscillation patterns under elevated temperature could inform future research on the development of strategies to mitigate the impacts of elevated temperatures on coral reefs.

## Methods

### Coral collection and experimental design

Twenty mature colonies of the reef-building coral *A. tenuis* (measuring 10–20 cm) were collected from a depth of 3- to 4-m on a tropical coral reef in Sanya (109° 29' E, 18° 12' N), Hainan Island, China. The colonies were fragmented into branches (measuring height ~5 cm, total of ~212 branches). The branches were then placed in a 500-L outdoor tank under natural condition for an acclimatization period of 20 days. The outdoor tank (open system) supplied with seawater collected from the Sanya reef at a depth of 3- to 4-m. After the acclimation period under natural condition, the branches were randomly placed into 9 indoor tanks (150-L) and cultivated under a simulated LD cycle (i.e., sunrise at 06:00 and sunset at 18:00 on November 6, 2021; light intensity of $20 \pm 5$ to $200 \pm 10$ μmol photons m$^{-2}$ s$^{-1}$ during daytime and seawater temperature of $27 \pm 0.5$ °C during 7 days for entrainment). Subsequently, branches were divided into three experimental subgroups: corals cultivated under LD ($27 \pm 0.5$ °C and 12:12 light:dark cycle), DD ($27 \pm 0.5$ °C and 0:24 light:dark cycle) and HLD conditions ($32 \pm 0.5$ °C and 12:12 light: dark cycle). Temperature conditions reflecting the annual mean temperature (27 °C; control) or elevated summer temperature (32 °C; heat stress) in Sanya reef.

Sampling began at night (i.e., time 0 = 24:00) of 13 November 2021 and was performed at 6-h intervals over 48–72 h. The indoor tanks underwent a daily seawater renewal rate of 25% using pre-warmed in-situ seawater.

All indoor tanks were filled with seawater collected from Sanya, with a pH of $8.15 \pm 0.003$, a salinity of $33.50 \pm 0.015$, a $NH_4^+$ concentration of $10 \pm 2$ μg/L, a $NO_3^-$ concentration of $32 \pm 2$ μg/L, a $PO_4^{3-}$ concentration of $9 \pm 1$ μg/L. The light and temperature of the tanks were generated and maintained according to the methods presented by Gong et al.[43]. The methods of monitor temperature, salinity, pH and inorganic nutrients (ammonia, nitrate, nitrite and phosphate) in the cultivation systems were recorded according to Gong et al.[43]. The occurrence of visible coral bleaching was recorded through snapshots taken by a GoPro (HERO9 Black, USA).

Spectral characteristics of coral growth sites and study site (depth of 3–4 m) were observed using Diving-PAM-II (Walz, Germany).

### Symbiodiniaceae cell density and maximal PSII quantum yield ($F_V/F_M$)

At each time point, nine coral branches were used to monitor Symbiodiniaceae cell density and $F_V/F_M$. To measure Symbiodiniaceae cell density, coral tissue was removed using a Waterpik containing filtered seawater (0.45 μm). The initial volume of the resulting slurry was measured with a graduated cylinder. The slurry was homogenized by votex and subsampled into four 3-ml aliquots. Subsamples were centrifuged (6500 r/min) for 5 min. After discarding the supernatant, the pellet containing algal cells was preserved in 1 ml of 5% formaldehyde at 4 °C for further analysis. The cell density of Symbiodiniaceae was counted using a hemocytometer counts under a microscope (CX21, Olympus, Japan). Density of Symbiodiniaceae cells was normalized to coral surface area by correlation between weight and surface area of aluminum foil imprints[44].

The highest photosystem II (PS II) photochemical quantum yield in coral holobiont, known as $F_V/F_M$ value, was assessed using pulse-amplitude modulation fluorometry (Diving-PAM-II, Walz, Germany) according to our previous study[43].

### Measurement of the metabolitic features of coral holobiont

The production of ROS by Symbiodiniaceae cells in *A. tenuis* was qualified according to a recent study[45]. For this, we first obtained freshly isolated Symbiodiniaceae cells in *A. tenuis* according to the above method. Then, the pellet containing freshly isolated Symbiodiniaceae cells was placed in 1 mL Eppendorf tubes and incubated under different conditions (similar with each time point for sampling) for 30 min. To measure ROS production, 5 μM CellROX orange (Life Technologies) was added to cultivated cells. Subsequently, cells were removed by centrifugation and the remaining supernatant (two-hundred-microliter aliquots) was immediately transferred to a 96-well plate and incubated in the dark at 37 °C for another 30 min. The fluorescence intensity of CellROX was quantified using a SpectraMax to calculate the relative release of ROS.

To assess the level of oxidation stress in the tissues of the host, we conducted an analysis of lipid peroxidation by quantifying the total malondialdehyde content[45]. The malondialdehyde content was then normalized to coral surface area.

### RNA extraction, sequencing and metatranscriptomic analysis

At each time point, total RNA was extract from three coral branches following the protocol provided by the Qiagen RNeasy Kit (Qiagen, Hilden, Germany). The quality and quantity of extracted RNA were assessed using Agilent 2100 Bioanalyzer (Santa Clara, CA, USA) and NanoDrop ND-1000 spectrometer (Wilmington, DE, USA). RNA samples with OD260/280 values ranging from 1.8 to 2.1 were selected for further cDNA library construction. The raw RNA sequences were analyzed by using the SqueezeMeta software, an automatic pipeline for metagenomics/metatranscriptomics data sets[46]. The LCA algorithm in SqueezeMeta assigned taxonomic information to the assembled genes by identifying the last common ancestor hit through a Diamond search against the GenBank nr database (included the annotated genomes of Symbiodiniaceae and *Acropora*)[46]. The software Bowtie2[47] was used to align original reads to contigs resulting from assembly to estimate the abundance of assembled genes in different samples. RSEM software[48] computed average coverage and normalized TMP values.

### Rhythmicity analysis of physiology and metabolism and gene transcription

Rhythmicity in physiology and metabolism and gene transcription was identified using DiscoRhythm (an online tool to explore cyclical temporal data)[49]. For diel gene transcription analyses, genes with a *p* value < 0.02 were considered confident cyclers. To perform clustering of the rhythmic transcripts based on GMMs, we used the Mclust function on the R mclust

package (V.5.2). The analyses were run separately for the 24-h oscillating genes from coral host or its hosted Symbiodiniaceae. Heatmaps were generated using the heatmap.2 function on the R package gplots (V.2.17.0). Venn diagrams were generated using the web tool Venn diagram (http://bioinformatics.psb.ugent.be/webtools/Venn/).

### Reporting summary

Further information on research design is available in the Nature Portfolio Reporting Summary linked to this article.

### Data availability

The raw sequence data of RNA sequencing libraries generated during the current study were deposited in the Sequence Read Archive (PRJNA1026720) of the NCBI. All data generated during this study are included in this published article and its Supplementary Data 1-3.

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

## Acknowledgements

This work was supported by the Natural Science Foundation of Guang Dong (2022A1515010521). The opening Project of Guangxi Laboratory on the Study of Coral Reefs in the South China Sea, Nanning 530004, China (GXLSCRSCS2019003).

## Author contributions

All authors contribute to the preparation of the previous and revised manuscript, which can be seen as follows: Sanqiang Gong: Conceptualization, Investigation, Sampling, Methodology, Formal analysis, Writing-original draft, and Funding acquisition. Jiayuan Liang, Investigation, Writing-review and editing. Lijia Xu, Resources, Writing-review and editing. Yongzhi Wang, sampling and data collection. Jun Li, Resources, Writing-review and editing. Xuejie, Jin, Writing-review and editing. Kefu Yu, Resources, Writing-review and editing and Funding acquisition. Yuehuan Zhang, Resources, Writing-review and editing.

## Competing interests

The authors declare no competing interests.
