## [Peer Review File · Communications Biology]

Reviewers' comments:

Reviewer #1 (Remarks to the Author):

I did not review the entire manuscript, as there are key problems in just the title and the abstract- though I did also look at the figures and methods. The manuscript is not well written and not of a standard for publication. I suggest the authors collaborate with individuals experienced at writing scientific papers before resubmission. Examples of statements that are simply not appropriate or accurate from just the title and abstract include:

The title... "Ocean warming-entrainable...."

The authors do not study ocean warming at all. They compare two laboratory tank temperatures. These are not the same thing at all. As noted below, one is really a tank heat shock more than anything. The title must reflect what your experiments showed, not what you conjecture the results meaning in the ocean ecosystem

p2L23. "Our aim was to elucidate how these two biological clocks respond to the elevated temperature". What two biological clocks? You have not mentioned any yet.

p2L26 "with the lowest values occurring at dusk". Lowest values of what? You just mentioned three different variables. All three? The writing does not explain what you mean.

p2L27 "However, Fv/Fm (photosystem II efficiency) showed the lowest oscillation level at midday, indicating that the rhythmic expulsion of algal cells causing coral bleaching was consistent with ROS leakage, but occurred after the photoinhibition of algal symbiont. "

I'm afraid this sentence makes little sense. Is this HLD? Do you mean LD vs HLD?

Terminology. An LD cycle is a light (L) dark (D) cycle, not a day - night cycle. In the lab, as the authors are, they are dealing with light-dark cycles under constant temperatures. This is very different to a day-night cycle in a natural environment. The language should be precise and state what was actually the case.

In general the term circadian is used when the authors really mean diel.

The statement that the 'findings support the presence of endogenous clocks in coral hosts' is hardly new, papers showing this have been available for well over a decade- what is the novel observation? The abstract should highlight what is new in the results.

I suggest a complete rewrite with additional writing input. There is some useful data buried here, but the writing makes it unusable for any scientific journal in the present state.

Reviewer #2 (Remarks to the Author):

This article investigates an interesting topic at the nexus of coral bleaching, global warming and holobiont chronobiology.

The main contribution of this paper is to track the metabolic and rhythmic transcriptional changes while the coral host is bleaching. They managed to distinguish coral transcriptome from the symbiont to identify rhythmic genes in their different experimental conditions (LD, DD, HLD). Their experiments showed that the host circadian clock is temperature compensated, while the arrhythmic Symbiodiniaceae transcriptome showed interesting neo-oscillations in response to temperature increase (in LD).

It's a fascinating topic for marine chronobiologists to interrogate the function of the circadian clock in such context because it questions if and how biological oscillations could be structural to the holobiont integrity. But, it is technically challenging to experimentally address the causality between the holobiont oscillation and the symbiosis during bleaching process. Authors choose an experimental design in free-running (DD) which surely underestimates the number of rhythmic in genes: the clock controlled genes. However, the LD and HLD experimental designs (Metabolic and Transcriptomic) are excellent and provide interesting correlative data and establish interesting ground for further studies. Indeed, for now it is still impossible to know if the oscillatory changes are the origin of the bleaching, or the consequences or even simply correlatives.

The interpretation of the data by the authors left me a bit confused even after reading multiples times the manuscript. And I would say that it's because there are some confusions with the choice of the specific terms of the chronobiology field, please see my following remarks below.

1_General and major comment about the rhythmic/circadian interpretation:

I am not sure we agree about the different terms used in chronobiology.

Rhythmic: any significant rhythm.

Circadian/diel: 24h-rhythm. (rhythmic in LD)

Circadian clock controlled: a 24h-rhythm maintained in absence of external cues. (rhythmic in DD)

The term "Circadian" used alone is confusing because the etymology means simply 24h rhythmic but is often used to mean "Circadian clock-controlled". I would recommend avoiding this formulation. Please be clearer for the reader.

Furthermore, any data would be considered as circadian clock-controlled if and only if these data are significantly rhythmic in free-running conditions (DD or LL). If it is rhythmic in LD, or HLD it cannot be considered as circadian clock-controlled. But simply as a diel rhythmicity, that could be driven by direct external sensory input (Light and temperature).

2_Results comments:

a_Physiology and metabolism rhythmic of coral holobiont

>Use algorithms (RAIN, LSP, COSINOR, you have many choices, you can even use online tools such as DiscoRhythm) to analyze rhythmicity of your metabolic and physiological data. Similar approach as you did for the following transcriptome. It will strengthen your point if you are consistent with your data analysis along the paper.

>Fig.1 : I understand that you can't use ZT in the x-axis because you mix different light conditions, but adding visually Day and Night (as Fig.2) will help to see the data along the time of the day. Be consistent in the graphic design between figures.

b_Transcriptome rhythmic analysis of coral holobiont

>Add in supplementary data the RAIN output.

c_Diel gene expression of coral holobiont

>Fig.2B: Cluster1 of the Symbiodiniaceae looks almost like a 12-15h period, no?

>Fig.2ABCD: Add the A and B the same color code as in C and D to help reader to identify the temporal pattern of each clusters.

>Fig.3A : The visual signal for Day and Night on X-axis is switched from what I understood in LD and HLD. It should be Day – Night etc...otherwise it's not coherent with your Fig.2A.

>Fig.3E: Please elaborate on the KEGG pathway enrichment you performed, which p-value you used. There is no mention of this analysis on the Methods, only a few words in the legend. I see the legend indicates 2500 genes. Did the authors perform the analysis on the total transcriptome and not only rhythmic genes?

d_Rhythmic genes in host and their oscillation under heat stress

>Fig.4: Separate potential oscillator's genes from the clock-controlled genes (CCG), on the figure. It is confusing to have them altogether without a clear visual signal to distinguish them.

e_Rhythmic genes in symbiont driven by heat stress.

>Fig.5: What authors defined as G-index seems to give similar information as relative amplitude. What does G stand for?

3_Discussion comments:

>Line 267-268: "Additionally, we identified for the first time several putative core circadian clock genes, including GAK, PNPO, PRODH, HEY and DNCL2 » .

Do authors mean clock-controlled genes?

>Line 302-307: "These genes, including HPH, DHODH, GPD1, rpoC, DNM1L, MAPK1_3, MPV17, RTCB, ALKBH2 and DNAJB8, consistently exhibit 24-hour rhythmic oscillation under heat stress, with peak expression at dusk, suggesting valid circadian regulation. In contrast, these genes showed a weak or a inconsistent circadian rhythm under LD or DD, suggesting that they could be involved in temperature-dependent circadian regulation."

If gene A is rhythmic in HLD, but not (weak or inconsistent) under LD and especially not in DD is strongly support that gene A is temperature sensitive and light dependent. But not under circadian clock regulation.

>Line 363-364: "These findings suggest that corals may be able to adapt to different climate change scenarios by altering their day-night cycles."

Do Authors imply the coral bleaching would be an adaptation to climate change? Can you elaborate on that?

Reviewer #3 (Remarks to the Author):

This study provides valuable insights into the rhythmic oscillations of the physiology, metabolism, and metatranscriptome of the coral holobiont under different conditions. The findings suggest the presence of endogenous clocks in the coral host and highlight the different temperature compensation mechanisms of rhythmic gene transcription in the coral and its symbiotic Symbiodiniaceae. Specifically, this study revealed the amplified rhythmic oscillation of Symbiodiniaceae is a key factor contributing to coral bleaching under elevated temperature conditions. The study design and data analysis are robust, and the results are well-supported by the data presented. Overall, I find the study to be interesting and potentially valuable to the field, and I suggest to be accept this article after some minor revisions

1. line 48: an extra space should be deleted.

2. line 55-57: the molecular processes involved in symbiosis between coral and Symbiodiniaceae have been extensively studied, I guess that the authors want to express that “studies about molecular processes related to rhythm of coral holobiont is limited”, please revise the expression more accurately.

3. line 102-104: light/dark cycle at 27 °C-LD; constant darkness at 27 °C-DD; light/dark cycle at 32 °C-HLD should be revised into LD, DD, and HLD to avoid duplication.

4. line 110: Please delete “However”.

5. line 163-171: please change the following long sentences to short sentences “Interestingly, most of these genes cycle under LD, indicating that their rhythmicity is driven by light. Furthermore, exposure to HLD resulted in the loss of rhythmicity in most genes (n = 517) that were rhythmic under LD, suggesting that heat stress disrupted the diel oscillation of host gene transcription. For the symbiont, we identified 17, 6, and 124 genes that displayed 24-hour oscillations under LD, DD, and HLD conditions, respectively (RAIN $P < 0.01$; Fig. 2D). Notably, most of genes of the symbiont (n = 124) showed rhythmicity under HLD, indicating that their rhythmic expression was primarily influenced by temperature.”

6. line 190: please change “Among the rhythmic genes” to “Among the rhythmic genes in host”.

7. line 191-196: These genes, i.e., HSP90B (heat shock protein 90kDa), GAK (cyclin G-associated kinase), PNPO (pyridoxamine 5'-phosphate oxidase), PRODH (proline dehydrogenase), HEY (hairy and enhancer of split-related with YRPW), DNCL2 (dynein light chain), and protein mab21-like 2, are not typical clock genes. So I suggest to revised related expression to “several putative core genes involved in circadian clock”.

8. line 201: please revised related expression according to the above suggestion.
9. line 214-215: please change “the mean expression level of” to “the mean expression level of rhythmic genes”.
10. line 265: change “key clock genes” to “core rhythmic genes”.
11. line 267-268: please revised related expression according to the above suggestion (7).
12. line 301: please delete “ indicating temperature-dependent circadian regulation”, as this description is duplicated from the previous expression in line 298-299.
13. line 330: to change “bleaching” to “The phenomenon of coral bleaching”. 14. line 649: “collected” should be revised into “cultivated or cultured”.

In conclusion, I believe that this study has the potential to make a valuable contribution to the field. However, I recommend that the authors address the concerns and suggestions outlined above before considering the manuscript for publication. With these revisions, I believe that the study will significantly enhance our understanding of the topic and be of interest to the readers of Communications Biology.

Corrections to the manuscript

Dear Reviewers,

We appreciate the time and efforts by the reviewers in reviewing the manuscript entitled “Ocean warming-entrainable circadian clocks in coral-Symbiodiniaceae holobiont”(Manuscript ID: COMMSBIO-23-4034A). The title has been revised into “Diel transcriptional response of coral-Symbiodiniaceae holobiont to elevated temperature” in revised manuscript. The newly submitted manuscript has made point-to-point revision according to the reviewer’s comments. In the revised manuscript, all corrections have been marked in red. The detailed responses and corrections are listed below.

Reviewer #1

1. I did not review the entire manuscript, as there are key problems in just the title and the abstract- though I did also look at the figures and methods. The manuscript is not well written and not of a standard for publication. I suggest the authors collaborate with individuals experienced at writing scientific papers before resubmission. Examples of statements that are simply not appropriate or accurate from just the title and abstract include:

The title... "Ocean warming-entrainable...."

The authors do not study ocean warming at all. They compare two laboratory tank temperatures. These are not the same thing at all. As noted below, one is really a tank heat shock more than anything. The title must reflect what your experiments showed,

not what you conjecture the results meaning in the ocean ecosystem

Response: Thanks for the meticulous checking and helpful suggestions. The title of this manuscript has been revised into “Diel transcriptional response of coral-Symbiodiniaceae holobiont to elevated temperature”. The abstract, introduction, results, discussion parts of this manuscript have been rewritten and revised according to the style of scientific papers.

2. p2L23. “Our aim was to elucidate how these two biological clocks respond to the elevated temperature”. What two biological clocks? You have not mentioned any yet.

Response: Related sentences , “In this study, we investigated the rhythmic oscillations of the physiology, metabolism, and metatranscriptome of *Acropora tenuis-Cladocopium* sp. holobiont under different conditions: day-night cycle (LD), constant darkness (DD), and day-night cycle with elevated temperature (HLD). Our aim was to elucidate how these two biological clocks respond to the elevated temperature, which leads to coral bleaching.”, have been revised into “Coral exhibits diel rhythms in behavior and gene transcription. However, the influence of elevated temperature, a key factor causing coral bleaching, on these rhythms remains poorly understood.” in revised manuscript.

3. p2L26 “with the lowest values occurring at dusk”. Lowest values of what? You just mentioned three different variables. All three? The writing does not explain what you mean.

Response: related description has been changed into “Under LD, the values of photosystem II efficiency, reactive oxygen species leakage, and lipid peroxidation

exhibited significant diel oscillations. These oscillations were further amplified during coral bleaching under HLD.”

4. p2L27 “However, F_v/F_m (photosystem II efficiency) showed the lowest oscillation level at midday, indicating that the rhythmic expulsion of algal cells causing coral bleaching was consistent with ROS leakage, but occurred after the photoinhibition of algal symbiont. “I’m afraid this sentence makes little sense. Is this HLD? Do you mean LD vs HLD?”

Response: We detected rhythmicity of the values of photosystem II efficiency, reactive oxygen species leakage, and lipid peroxidation, and also algal cell density using the method suggested by another reviewer, and found the diel oscillation of algal cell density was not significant. So, related results have been revised in newly revised manuscript. In old version of manuscript, we mean in HLD, not mean LD vs HLD. In discussion part of newly revised manuscript, we have added related description and discussion “We observed that the F_v/F_m exhibited its lowest oscillation value at midday, while ROS leakage from isolated algal symbionts and lipid peroxidation levels in the host peaked at dusk under elevated temperature. This suggests that increased oxidative damage to the coral host resulting from ROS leakage lags behind photosynthetic inhibition in symbiont.

5. Terminology. An LD cycle is a light (L) dark (D) cycle, not a day - night cycle. In the lab, as the authors are, they are dealing with light-dark cycles under constant temperatures. This is very different to a day-night cycle in a natural environment. The language should be precise and state what was actually the case.

In general the term circadian is used when the authors really mean diel.

Response: We agree with your suggestions. In newly revised manuscript, day-night cycle has been changed into light-dark cycle, and “circadian” has been changed into “diel” in some sentences.

6. The statement that the ‘findings support the presence of endogenous clocks in coral hosts’ is hardly new, papers showing this have been available for well over a decade- what is the novel observation? The abstract should highlight what is new in the results.

Response: We agree with your suggestions. the abstract has been revised according to your suggestions and our present results. “Coral exhibits diel rhythms in behavior and gene transcription. However, the influence of elevated temperature, a key factor causing coral bleaching, on these rhythms remains poorly understood. To address this, we examined physiological, metabolic, and gene transcription oscillations in the *Acropora tenuis-Cladocopium* sp. holobiont under constant darkness (DD), light-dark cycle (LD), and LD with elevated temperature (HLD). Under LD, the values of photosystem II efficiency, reactive oxygen species leakage, and lipid peroxidation exhibited significant diel oscillations. These oscillations were further amplified during coral bleaching under HLD. Gene transcription analysis identified 24-hour rhythms for specific genes in both coral and Symbiodiniaceae under LD. Notably, these rhythms were disrupted in coral and shifted in Symbiodiniaceae under HLD. Importantly, we identified over 20 core rhythmic genes in coral-Symbiodiniaceae holobiont, including clock genes. Specifically, *CIPC* (CLOCK-interacting pacemaker-like) gene was

suggested as a core clock gene in *A. tenuis*. Elevated temperature significantly down-regulated the transcription of two abundant rhythmic genes encoding glycoside hydrolases (*CBM21*) and heme-binding protein (*SOUL*). These findings indicate that elevated temperatures disrupt diel gene transcription rhythms in the coral-Symbiodiniaceae holobiont, affecting essential processes such as carbohydrate utilization and redox homeostasis. These disruptions may contribute to the thermal bleaching observed in coral.”.

7. I suggest a complete rewrite with additional writing input. There is some useful data buried here, but the writing makes it unusable for any scientific journal in the present state.

Response: thanks for your good suggestions. The abstract, introduction, results, discussion parts of this manuscript have been rewritten and revised according to the style of scientific papers. In addition, another 24 hours data under DD conditions and more replicates for monitor F_v/F_m , ROS, host lipid peroxidation and algal density were added in newly revised manuscript.

Thanks for the meticulous checking and helpful suggestions again.

Reviewer #2:

This article investigates an interesting topic at the nexus of coral bleaching, global warming and holobiont chronobiology.

The main contribution of this paper is to track the metabolic and rhythmic transcriptional changes while the coral host is bleaching. They managed to distinguish

coral transcriptome from the symbiont to identify rhythmic genes in their different experimental conditions (LD, DD, HLD). Their experiments showed that the host circadian clock is temperature compensated, while the arrhythmic Symbiodiniaceae transcriptome showed interesting neo-oscillations in response to temperature increase (in LD).

It's a fascinating topic for marine chronobiologists to interrogate the function of the circadian clock in such context because it questions if and how biological oscillations could be structural to the holobiont integrity. But, it is technically challenging to experimentally address the causality between the holobiont oscillation and the symbiosis during bleaching process. Authors choose an experimental design in free-running (DD) which surely underestimates the number of rhythmic in genes: the clock controlled genes. However, the LD and HLD experimental designs (Metabolic and Transcriptomic) are excellent and provide interesting correlative data and establish interesting ground for further studies. Indeed, for now it is still impossible to know if the oscillatory changes are the origin of the bleaching, or the consequences or even simply correlatives.

The interpretation of the data by the authors left me a bit confused even after reading multiples times the manuscript. And I would say that it's because there are some confusions with the choice of the specific terms of the chronobiology field, please see my following remarks below.

Response: Thanks for the meticulous checking and helpful suggestions. Firstly, the abstract, introduction, results, discussion parts of this manuscript have been rewritten

and revised according to the style of scientific papers. Secondly, another data (24 hours) under DD conditions and more replicates for monitor *Fv/Fm*, ROS, host lipid peroxidation and algal density were added in newly revised manuscript. Thirdly, we the oscillation patterns of host and its hosted Symbiodiniaceae were shown in results parts, separately. The overall oscillation patterns based on diel rhythmic genes (identified on all conditions were included, DD, LD, and HLD) were exhibited in newly revised manuscript, and we found different oscillation patterns of host and its hosted Symbiodiniaceae under both LD and HLD conditions. These new insights are buried in old version of manuscript. Then, we analyzed the core rhythmic genes (including clock genes).

1_General and major comment about the rhythmic/circadian interpretation:

I am not sure we agree about the different terms used in chronobiology.

Rhythmic: any significant rhythm.

Circadian/diel: 24h-rhythm. (rhythmic in LD)

Circadian clock controlled: a 24h-rhythm maintained in absence of external cues.
(rhythmic in DD)

The term “Circadian” used alone is confusing because the etymology means simply 24h rhythmic but is often used to mean “Circadian clock-controlled”. I would recommend avoiding this formulation. Please be clearer for the reader.

Furthermore, any data would be considered as circadian clock-controlled if and only if these data are significantly rhythmic in free-running conditions (DD or LL). If it is rhythmic in LD, or HLD it cannot be considered as circadian clock-controlled. But

simply as a diel rhythmicity, that could be driven by direct external sensory input (Light and temperature).

Response: Thanks for your experienced suggestions. I have read these suggestions many times.

- Rhythmic: means any significant rhythms, including characters of physiology, metabolism, and gene transcription.
- Circadian/diel: means 24h-rhythm. (rhythmic under LD)
- Circadian clock controlled: means a 24h-rhythm maintained in absence of external cues. (rhythmic in DD)

Related descriptions have been revised according to your suggestion in newly submitted manuscript.

2_Results comments:

2.1. a_Physiology and metabolism rhythmic of coral holobiont

>Use algorithms (RAIN, LSP, COSINOR, you have many choices, you can even use online tools such as DiscoRhythm) to analyze rhythmicity of your metabolic and physiological data. Similar approach as you did for the following transcriptome. It will strengthen your point if you are consistent with your data analysis along the paper.

>Fig.1 : I understand that you can't use ZT in the x-axis because you mix different light conditions, but adding visually Day and Night (as Fig.2) will help to see the data along the time of the day. Be consistent in the graphic design between figures.

Response: Thanks for your experienced suggestions.

- In newly revised manuscript, we have analyzed all data based on the online tools –

DiscoRhythm according to your suggestions.

- We have added visually boxes to mark light (day)-dark (night) cycles.

2.2. b_Transcriptome rhythmic analysis of coral holobiont

>Add in supplementary data the RAIN output.

Response: Supplementary files from DiscoRhythm have been added in newly revised manuscript.

2.3. c_Diel gene expression of coral holobiont

>Fig.2B: Cluster1 of the Symbiodiniaceae looks almost like a 12-15h period, no?

Response: Using DiscoRhythm, we also detected the main periods of both host and Symbiodiniaceae under DD, LD, and HLD, showing 24h period of all data sets. This differences may aroused by weak diel oscillation of the Symbiodiniaceae, but the main oscillations of Symbiodiniaceae under different conditions are 24h period.

2.4. >Fig.2ABCD: Add the A and B the same color code as in C and D to help reader to identify the temporal pattern of each clusters.

Response: all of the figures have been re-drawn according to your suggestions and the suggestions of other reviews.

2.5. >Fig.3A : The visual signal for Day and Night on X-axis is switched from what I understood in LD and HLD. It should be Day – Night etc...otherwise it's not coherent with your Fig.2A.

Response: Related figures has been changed to indicate light(day)-dark(night)cycle.

2.6. >Fig.3E: Please elaborate on the KEGG pathway enrichment you performed, which p-value you used. There is no mention of this analysis on the Methods, only a few words

in the legend. I see the legend indicates 2500 genes. Did the authors perform the analysis on the total transcriptome and not only rhythmic genes?

Response: In newly submitted manuscript, the figure of KEGG pathway has been deleted. In original manuscript, the 2500 represents the abundance of genes in KEGG pathway. In revised figures, figure2B, figure2E, and Figure5, we only focused on identified rhythmic genes (figure2B, figure2E) or core rhythmic genes (Figure5).

2.7. d_Rhythmic genes in host and their oscillation under heat stress

>Fig.4: Separate potential oscillator' s genes from the clock-controlled genes (CCG), on the figure.

It is confusing to have them altogether without a clear visual signal to distinguish them.

Response: In newly revised manuscript, the potential circadian clock genes were shown in Figure.3C. The other core genes (potential circadian clock-controlled genes) were shown in Figure 4 A (host) and 4B (symbiont)

2.8. e_Rhythmic genes in symbiont driven by heat stress.

>Fig.5: What authors defined as G-index seems to give similar information as relative amplitude. What does G stand for?

Response: This figure has been deleted in newly submitted manuscript, and Figure 4B was added to show diel oscillation of core rhythmic genes identified in symbiont.

3_Discussion comments:

3.1. >Line 267-268: “Additionally, we identified for the first time several putative core circadian clock genes, including GAK, PNPO, PRODH, HEY and DNCL2 » .

Do authors mean clock-controlled genes?

Response: Possible, those genes were circadian clock-controlled genes. According to another reviewer's suggestion, the discussion of newly submitted manuscript has been revised into two part.

- The candidate circadian clock genes and their oscillation in coral host or symbiont under elevated temperature.
- Diel rhythmic oscillation of genes related to coral bleaching.

3.2. >Line 302-307: “These genes, including HPH, DHODH, GPD1, rpoC, DNMI1L, MAPK1_3, MPV17, RTCB, ALKBH2 and DNAJB8, consistently exhibit 24-hour rhythmic oscillation under heat stress, with peak expression at dusk, suggesting valid circadian regulation. In contrast, these genes showed a weak or a inconsistent circadian rhythm under LD or DD, suggesting that they could be involved in temperature-dependent circadian regulation.”

If gene A is rhythmic in HLD, but not (weak or inconsistent) under LD and especially not in DD is strongly support that gene A is temperature sensitive and light dependent. But not under circadian clock regulation.

Response: related descriptions have been changed throughout the manuscript according to your experienced suggestions.

>Line 363-364: “These findings suggest that corals may be able to adapt to different climate change scenarios by altering their day-night cycles.”

Do Authors imply the coral bleaching would be an adaptation to climate change? Can you elaborate on that?

Response: The discussion has been majorly revised into two different part as say above.

And related discussion about adaptation of coral to climate change has been deleted in newly submitted manuscript.

Reviewer #3 (Remarks to the Author):

This study provides valuable insights into the rhythmic oscillations of the physiology, metabolism, and metatranscriptome of the coral holobiont under different conditions. The findings suggest the presence of endogenous clocks in the coral host and highlight the different temperature compensation mechanisms of rhythmic gene transcription in the coral and its symbiotic Symbiodiniaceae. Specifically, this study revealed the amplified rhythmic oscillation of Symbiodiniaceae is a key factor contributing to coral bleaching under elevated temperature conditions. The study design and data analysis are robust, and the results are well-supported by the data presented. Overall, I find the study to be interesting and potentially valuable to the field, and I suggest to be accept this article after some minor revisions

Thanks for the meticulous checking and helpful suggestions again. Notably, the abstract, introduction, results, discussion parts of this manuscript have been rewritten and revised according to the style of scientific papers. Add new data sets have been added in newly revised manuscript.

1. line 48: an extra space should be deleted.

Response: we have checked related style of written throughout the newly submitted manuscript, and all extra spaces have been deleted.

2. line 55-57: the molecular processes involved in symbiosis between coral and

Symbiodiniaceae have been extensively studied, I guess that the authors want to express that “studies about molecular processes related to rhythm of coral holobiont is limited”, please revise the expression more accurately.

Response: related description has been revised in newly submitted manuscript.

3. line 102-104: light/dark cycle at 27 °C-LD; constant darkness at 27 °C-DD; light/dark cycle at 32 °C-HLD should be revised into LD, DD, and HLD to avoid duplication.

Response: we have checked related style of written throughout the revised manuscript, and related words have been changed to avoid duplication according to your suggestion.

4. line 110: Please delete “However”.

Response: “However” has been deleted in newly revised manuscript.

5. line 163-171: please change the following long sentences to short sentences “Interestingly, most of these genes cycle under LD, indicating that their rhythmicity is driven by light. Furthermore, exposure to HLD resulted in the loss of rhythmicity in most genes (n = 517) that were rhythmic under LD, suggesting that heat stress disrupted the diel oscillation of host gene transcription. For the symbiont, we identified 17, 6, and 124 genes that displayed 24-hour oscillations under LD, DD, and HLD conditions, respectively (RAIN $P < 0.01$; Fig. 2D). Notably, most of genes of the symbiont (n = 124) showed rhythmicity under HLD, indicating that their rhythmic expression was primarily influenced by temperature.”

Response: Related sentences have been changed into “Under DD, LD, and HLD conditions, 100, 522, and 72 rhythmic genes were identified in the host, respectively (based on Cosinor and JTK_CYCLE algorithms, $p < 0.02$, Fig. 2A, supplementary file

1). In contrast, 75, 64, and 93 rhythmic genes were detected in the symbiont (based on Cosinor and JTK_CYCLE algorithms, $p < 0.02$, Fig. 2D, supplementary file 2).

Among the identified rhythmic genes, 25 exhibited oscillatory transcription in the host under both DD and LD conditions, designated as core rhythmic genes (Fig. 2A, supplementary file 3).” in revised manuscript according to your suggestions and the suggestions from another reviewer.

6. line 190: please change “Among the rhythmic genes” to “Among the rhythmic genes in host”.

Response: Related expression has been revised according to your suggestion.

7. line 191-196: These genes, i.e., HSP90B (heat shock protein 90kDa), GAK (cyclin G-associated kinase), PNPO (pyridoxamine 5'-phosphate oxidase), PRODH (proline dehydrogenase), HEY (hairy and enhancer of split-related with YRPW), DNCL2 (dynein light chain), and protein mab21-like 2, are not typical clock genes. So I suggest to revised related expression to “several putative core genes involved in circadian clock”.

Response: In revised manuscript, core rhythmic genes have been found to be clock gene in other model animals were described as candidate circadian clock genes in coral or Symbiodiniaceae. The other core rhythmic genes significant oscillation under both DD and LD condition were suggested as candidate circadian clock-controlled genes in revised manuscript.

8. line 201: please revised related expression according to the above suggestion.

Response: related descriptions of circadian, clock gene or clock-controlled genes have

been check throughout the manuscript, and revised according to your suggestions and the suggestions from other reviewers.

9. line 214-215: please change “the mean expression level of” to “the mean expression level of rhythmic genes”.

Response: related description has been revised according to your suggestions.

10. line 265: change “key clock genes” to “core rhythmic genes”.

Response: related description has been revised according to your suggestions.

11. line 267-268: please revised related expression according to the above suggestion

Response: related description has been revised according to your suggestions.

12. line 301: please delete “ indicating temperature-dependent circadian regulation”, as this description is duplicated from the previous expression in line 298-299.

Response: related description has been deleted in revised manuscript.

13. line 330: to change “bleaching” to “The phenomenon of coral bleaching”. 14. line 649: “collected” should be revised into “cultivated or cultured”.

In conclusion, I believe that this study has the potential to make a valuable contribution to the field. However, I recommend that the authors address the concerns and suggestions outlined above before considering the manuscript for publication. With these revisions, I believe that the study will significantly enhance our understanding of the topic and be of interest to the readers of Communications Biology.

Response: related description has been revised in newly submitted manuscript.

Thanks for the meticulous checking and helpful suggestions.

Looking forward to hearing from you and with best wishes.

Sincerely yours,

Prof. Dr. Kefu Yu

Prof. Dr. Yuehuan Zhang

Dr. Sanqiang Gong

Reviewers' comments:

Reviewer #2 (Remarks to the Author):

Brief summary: This article investigates an interesting topic at the nexus of coral bleaching, global warming and holobiont chronobiology. The main contribution of this paper is to track the metabolic and rhythmic transcriptional changes while the coral host is bleaching. They managed to distinguish coral transcriptome from the symbiont to identify rhythmic genes in their different experimental conditions (LD, DD, HLD). They managed to identify core pacemaker genes in the host but not in the symbiont. Furthermore, they identified numerous rhythmic genes in both organism probably involved in the symbiosis which are dysregulated by the increased temperature condition (HLD)

Overall impression of the work: Following reviewers' recommendations, authors made great effort to improve the clarity of the manuscript and strengthen their conclusions with supplementary data in DD and rhythmic analysis. The current revised version still needs some work to reach its best intrinsic value. I provided here numerous recommendations to the authors for a second round of revision. Regarding the figures specifically, I would recommend to the authors to be very precise about the result they want to underscore when they design the figure. Please, keep in mind that some results can be moved to supplementary figures, if informative but not necessarily supporting any important point of your study.

Specific comments and recommendations:

Introduction:

1.Lines 60-63 : Generally unclear. This statement is not true as the "presence" of endogenous clocks in coral and Symbiodiniaceae is already known by the observation of rhythmic processes (example: Sorek, 2013, PMC3619499). You are addressing here important gap in the field, which I wrote as questions.

- First gap: What are the molecular factors of their respective biological clocks?
- Second gap: What are the mechanisms which underly their synchronized rhythms?
- Third gap: Does this synchronization facilitate the symbiosis?

However, in lines 69-71: you define your main gap to fill as: How elevated temperature influence rhythmic gene transcription which "potentially" regulate symbiosis during coral bleaching?

Overall, this is a bit confusing for the reader and to me as it seems you aim to address all in one.

I recommend to the authors to think carefully about the question they are addressing in their paper as it will generate the frame of your study and how you interpret your result and how you discuss your results.

2.Supplementary tables: Please add a first tab in each suppl. Table indicating a summary of the content of each supplementary table. Furthermore, the authors used a color coding which does not refer to anything. Please add a legend if the color coding is necessary or remove it.

Results:

Physiology and metabolism rhythmic of coral holobiont.

3.Line 85: remove "controlled".

4.Line 84 – 96: Please provide the output of the Discorhythm analysis of all the physiological and metabolic parameters you tested in a supplementary table. You can show only the Cosinor as you selected this one. But it would add more transparency to your analysis.

Furthermore, I would advise to generate a table 1 containing each parameter in each condition the corresponding p-value. This will directly support your analysis on the Fig1. As for now it is unclear in your result paragraph what is significantly rhythmic or not.

Transcriptome rhythmic analysis of coral holobiont.

General comment to write this paragraph: Basically, you have a phenotype in fig.1 and with your RNAseq study you are looking for the underlying cause of this phenotype. You should keep this in mind in the way you structured your results, figures and discussions.

5.Line 97 – 106: Please provide in supplementary table 1 the rhythmic analysis output for each condition. From what I understood by exploring the excel file, the p-value correspond only to the LD condition.

In general, you used the p-value instead of the q-value which is the multiple hypothesis testing to define rhythmic genes. To not use multiple hypothesis testing is increasing by default the number of false-positive rhythmic genes. However, it is difficult to identify many rhythmic genes passing through this threshold in cnidarian. You should write in your result paragraph how many genes you got with a $q.val < 0.05$ which are the “most” confident rhythmic genes.

– In the host dataset, with JTK in LD you mostly obtain putative core pacemaker genes with a $q.val < 0.05$, which is a good sign as they are often the most rhythmic genes in cnidarian. I cannot check for DD and HLD but you probably won't get a lot (if any) genes with this threshold.

I would proceed like this, go clear with your analysis. “We performed first a stringent analysis to identify rhythmic genes (JTK, $q.val < 0.05$) and obtain x genes in LD, y in DD and z in HLD. However, due to the difficulty to identify rhythmic genes except core pacemaker genes we opted for a $p < 0.02$, acknowledging the risk of false-positive...”. No need to generate another figure here.

Same comment for the Symbiodiniaceae analysis, where indeed no genes pass the $q.val < 0.05$ at least in LD. Please add the result for DD and HLD too in the supplementary table.

The oscillation of core rhythmic genes in coral holobiont under elevated temperature.

6.Line 124 and line 130: There is still a confusion between pacemaker genes (the circadian clock machinery, or molecular factors) and clock-controlled genes (genes under control of the pacemaker, which are not involved in the circadian oscillation). Please do not use the term “core rhythmic genes” as it confuses the reader.

Figure:

7.Figure 2:

7.1. in Fig2A and Fig2D, indicate if the data are from the host or from the symbiont, either with a scheme, or just write it above.

7.2. in Fig2B and Fig2E it is not obvious which data are shown on the heatmap. Be more explicit on the figure itself. As far as I understood you showed the LD rhythmic genes list (Cosinor or JTK?), in the three different conditions.

7.3. Fig.2C should appear with the clustering analysis we can see in Fig.3A. I would recommend to put them together in one single figure.

8.Figure 3: 1. This figure should only focus on core pacemaker genes of the host.

9.Figure 4: 1. What are you trying to say with this figure beside the fact you identified rhythmic gene

already shown before? Again, be visually explicit on the figure about the host vs. symbiont.

10. Figure 5: 1. Please differentiate the genes of the host from the symbiont. The idea is, I guess, to lead to the final summary figure 6. So, the design of this figure is very important.

11. Figure 6: Generating a summary figure is an excellent idea.

Discussion:

12. Line 210: what does mean: "specificity to particular cultivation conditions" ?

Reviewer #3 (Remarks to the Author):

I have thoroughly reviewed the revised manuscript titled "Diel transcriptional response of coral-Symbiodiniaceae holobiont to elevated temperature" submitted to Communications Biology. I appreciate the opportunity to review this work and provide my feedback.

The concerns and suggestions have been addressed in the newly submitted manuscript, and as a result, I recommend accepting this article for publication in Communications Biology.

Considering the significance of coral reef ecological restoration and protection, I believe that publishing this article in our journal would be highly valuable to the scientific community.

Sincerely

Corrections to the manuscript

Dear Reviewers,

We appreciate the time and efforts by the reviewers in reviewing the manuscript entitled **“Diel transcriptional responses of coral-Symbiodiniaceae holobiont to elevated temperature ”**(Manuscript ID: COMMSBIO-23-4034A). The newly submitted manuscript has made point-to-point revision according to the reviewer’s comments. In the revised manuscript, all corrections have been marked in red. The detailed responses and corrections are listed below.

Reviewer #2

1. Brief summary:

This article investigates an interesting topic at the nexus of coral bleaching, global warming and holobiont chronobiology. The main contribution of this paper is to track the metabolic and rhythmic transcriptional changes while the coral host is bleaching. They managed to distinguish coral transcriptome from the symbiont to identify rhythmic genes in their different experimental conditions (LD, DD, HLD). They managed to identify core pacemaker genes in the host but not in the symbiont. Furthermore, they identified numerous rhythmic genes in both organism probably involved in the symbiosis which are dysregulated by the increased temperature condition (HLD)

Overall impression of the work: Following reviewers’ recommendations, authors made great effort to improve the clarity of the manuscript and strengthen their conclusions

with supplementary data in DD and rhythmic analysis. The current revised version still needs some work to reach its best intrinsic value. I provided here numerous recommendations to the authors for a second round of revision. Regarding the figures specifically, I would recommend to the authors to be very precise about the result they want to underscore when they design the figure. Please, keep in mind that some results can be moved to supplementary figures, if informative but not necessarily supporting any important point of your study.

Response: Thanks for the meticulous checking and helpful suggestions. In all, the introduction, results and figures were revised according to your suggestions.

2. Introduction:

Lines 60-63 : Generally unclear. This statement is not true as the “presence” of endogenous clocks in coral and Symbiodiniaceae is already known by the observation of rhythmic processes (exemple: Sorek, 2013, PMC3619499). You are addressing here important gap in the field, which I wrote as questions.

- First gap: What are the molecular factors of their respective biological clocks?
- Second gap: What are the mechanisms which underly their synchronized rhythms?
- Third gap: Does this synchronization facilitate the symbiosis?

However, in lines 69-71: you define your main gap to fill as: How elevated temperature influence rhythmic gene transcription which “potentially” regulate symbiosis during coral bleaching?

Overall, this is a bit confusing for the reader and to me as it seems you aim to address all in one.

I recommend to the authors to think carefully about the question they are addressing in their paper as it will generate the frame of your study and how you interpret your result and how you discuss your results.

2. Supplementary tables: Please add a first tab in each suppl. Table indicating a summary of the content of each supplementary table. Furthermore, the authors used a color coding which does not refer to anything. Please add a legend if the color coding is necessary or remove it.

Response: We agree with your ideas of the present gaps in holobiont chronobiology of coral. And I (the first author of this manuscript) bear in mind your good questions: (1). What are the respective molecular factors of coral host and its Symbiodiniaceae? (2) What are the mechanisms which underly their synchronized rhythms? (3) Does this synchronization facilitate the symbiosis?.

The sentences in Lines 60-63 of raw manuscript have been revised into “Rhythmic processes were also observed in symbiotic Symbiodiniaceae, proposed the presence of endogenous clocks in both coral and its hosted Symbiodiniaceae.” in newly submitted manuscript. In revised manuscript, we only focus on gap of “How elevated temperature influence rhythmic gene transcription which “potentially” regulate symbiosis during coral bleaching?”.

For supplementary tables, a summary of the content of each supplementary table has been added in newly up-loaded supplementary fiels. In addition, color coding in table has been removed.

3. Results:

3.1. Line 85: remove “controlled”.

Response: the “controlled” has been deleted in revised manuscript.

3.2. Line 84 – 96: Please provide the output of the Discorhythm analysis of all the physiological and metabolic parameters you tested in a supplementary table. You can show only the Cosinor as you selected this one. But it would add more transparency to your analysis.

Furthermore, I would advise to generate a table 1 containing each parameter in each condition the corresponding p-value. This will directly support your analysis on the Fig1. As for now it is unclear in your result paragraph what is significantly rhythmic or not.

Response: 1. the output of the Discorhythm analysis of all the physiological and metabolic parameters was provided as a supplementary file1 in newly submitted manuscript. 2. A table 1 with corresponding p-value under different conditions were supplied according to your suggestion.

3.3. General comment to write this paragraphe: Basically, you have a phenotype in fig.1 and with your RNAseq study you are looking for the underlying cause of this phenotype. You should keep this in mind in the way you structured your results, figures and discussions.

Line 97 – 106: Please provide in supplementary table 1 the rhythmic analysis output for each condition. From what I understood by exploring the excel file, the p-value correspond only to the LD condition.

Response: the outputs of the rhythmic analysis of each conditions were provided in supplementary file 2 and supplementary file 3 in newly submitted manuscript according to your suggestions.

3.4. In general, you used the p-value instead of the q-value which is the multiple hypothesis testing to define rhythmic genes. To not use multiple hypothesis testing is increasing by default the number of false-positive rhythmic genes. However, it is difficult to identify many rhythmic genes passing through this threshold in cnidarian. You should write in your result paragraph how many genes you got with a $q.val < 0.05$ which are the “most” confident rhythmic genes.

In the host dataset, with JTK in LD you mostly obtain putative core pacemaker genes with a $q.val < 0.05$, which is a good sign as they are often the most rhythmic genes in cnidarian. I cannot check for DD and HLD but you probably won't get a lot (if any) genes with this threshold.

I would proceed like this, go clear with your analysis. “We performed first a stringent analysis to identify rhythmic genes (JTK, $q.val < 0.05$) and obtain x genes in LD, y in DD and z in HLD. However, due to the difficulty to identify rhythmic genes except core pacemaker genes we opted for a $p < 0.02$, acknowledging the risk of false-positive...”. No need to generate another figure here.

Same comment for the Symbiodiniaceae analysis, where indeed no genes pass the $q.val < 0.05$ at least in LD. Please add the result for DD and HLD too in the supplementary table.

Response: we are agree with your good suggestions. and related sentences have been

revised into “Firstly, a rigorous analysis was conducted to identify rhythmic genes (JTK + Cosinor, q-values < 0.05), resulting in 0, 34 and 0 genes identified in the coral host under DD, LD, and HLD conditions, respectively (supplementary file 2). For the symbiont, we found 0, 1 and 0 rhythmic genes under the same conditions (supplementary file 3). Considering the difficulty to identify rhythmic genes with q-values < 0.05, we then opted for p-values < 0.02 to identify rhythmic genes, acknowledging the risk for false-positive results. Under DD, LD, and HLD conditions, 100, 522, and 72 rhythmic genes were identified in the host, respectively (JTK + Cosinor, p-values < 0.02, Fig. 2A, supplementary file 2). In contrast, 75, 64, and 93 rhythmic genes were detected in the symbiont (JTK + Cosinor, p-values < 0.02, Fig. 2C, supplementary file 3). Among the identified rhythmic genes, 25 exhibited oscillatory transcription in the host under both DD and LD conditions. In the symbiont, we also detected several rhythmic genes, which exhibiting similar oscillations under both DD and LD conditions. These genes were designated as candidate clock or clock-controlled genes (supplementary file 4)” in newly submitted manuscript.

3.5. Line 124 and line 130: There is still a confusion between pacemaker genes (the circadian clock machinery, or molecular factors) and clock-controlled genes (genes under control of the pacemaker, which are not involved in the circadian oscillation). Please do not use the term “core rhythmic genes” as it confuses the reader.

Response: In newly submitted manuscript, we do not use the term “core rhythmic genes” as it confuses the readers. We use rhythmic genes, candidate clock genes or clock controlled genes in proper place.

4. Figure:

4.1. Figure 2: in Fig2A and Fig2D, indicate if the data are from the host or from the symbiont, either with a scheme, or just write it above.

Response: we have added host or symbiont above the figures in newly submitted manuscript.

4.2. in Fig2B and Fig2E it is not obvious which data are shown on the heatmap. Be more explicit on the figure itself. As far as I understood you showed the LD rhythmic genes list (Cosinor or JTK?), in the three different conditions.

Response: we have described that rhythmic genes identified from DD, LD and HLD conditions (JTK + Cosinor , p-values < 0.02, after deduplication) were shown in heatmaps in result of newly submitted manuscript. And in figure legends, we also added related description as follows: "Heatmaps depict the median gene expression values of rhythmic genes identified from DD, LD and HLD conditions (JTK + Cosinor , p-values < 0.02, after deduplication) over time points (n = 3 biological replicates), where each row represents a rhythmic gene in the host or symbiont.

4.3. Fig.2C should appear with the clustering analysis we can see in Fig.3A. I would recommend to put them together in one single figure.

Response: the two figures have been put together as a single figure in revised manuscript. and related descripton in original mansucrypt has been changed into "Furthermore, we observed that the overall oscillation patterns of rhythmic genes identified under LD (JTK + Cosinor , p-values < 0.02) were disrupted in the coral (Fig. 3A) and shifted in the symbiont under HLD (Fig. 3B). These disordered oscillation

patterns were further supported by Gaussian mixture model (GMM) (Fig. 3C and Fig. 3F) clustering of rhythmic genes identified under LD and HLD (JTK + Cosinor , p-values < 0.02), respectively. It was observed that most rhythmic genes (>90%) had high transcriptional abundance (peak) at dawn (6:00) or midday (12:00) in both the coral host and symbiont under LD. Conversely, under HLD, the patterns were altered, with 84% of rhythmic genes in the symbiont peaking at dusk (18:00).”

4.4. Figure 3: 1. This figure should only focus on core pacemaker genes of the host.

Response: this figure only focus on cadidate clock genes of host and as a new figure 4 in revised manuscript.

4.5. Figure 4: 1. What are you trying to say with this figure beside the fact you identified rhythmic gene already shown before? Again, be visually explicit on the figure about the host vs. symbiont.

Response: this figure focus on candidate clock controlled genes in host or symbiont. Due to duplication with other figures, we designated this figure as supplementary figure 1 (Fig. S1) in newly submitted manuscript and we also explicated on the figure about the host and symbiont.

4.6. Figure 5: 1. Please differentiate the genes of the host from the symbiont. The idea is, I guess, to lead to the final summary figure 6. So, the design of this figure is very important.

Response: we haved differentiated these genes by added host or symbiont in the right part of this figure. In addition, we also differentiated these genes as candidated clock genes or clock-controlled genes as this: “The capital letters of C and CCG represent

candidate clock gene, and clock-controlled gene, respectively”. The designation of these genes as candidate clock genes or clock-controlled genes were described in results as this “Among the identified rhythmic genes, 25 exhibited oscillatory transcription in the host under both DD and LD conditions. In the symbiont, we also detected several rhythmic genes, which exhibiting similar oscillations under both DD and LD conditions. These genes were designated as candidate clock or clock-controlled genes. In addition, we also shown two clock genes (*CLOCK* and *BMALI*), which have been widely reported in coral in this figure, as the two genes were also detected in our present study.

4.7 Figure 6: Generating a summary figure is an excellent idea.

Response: Thanks for the meticulous checking.

5. Discussion:

12.Line 210: what does mean: “specificity to particular cultivation conditions” ?

Response: We want to express “the majority of these rhythmic genes exhibited specificity to LD or HLD conditions, suggesting the transcription of these genes are majorly dependent on light-dark cycles or elevated temperature”. In newly revised manuscript, this sentence has been deleted due to uselessness for discussion.

Thanks again for the meticulous checking and helpful suggestions.

Reviewer #3 (Remarks to the Author):

I have thoroughly reviewed the revised manuscript titled "Diel transcriptional response of coral-Symbiodiniaceae holobiont to elevated temperature" submitted to Communications Biology. I appreciate the opportunity to review this work and provide

my feedback.

The concerns and suggestions have been addressed in the newly submitted manuscript, and as a result, I recommend accepting this article for publication in *Communications Biology*.

Considering the significance of coral reef ecological restoration and protection, I believe that publishing this article in our journal would be highly valuable to the scientific community.

Response: thanks for review our work and for your positive evaluation of our research.

Looking forward to hearing from you and with best wishes.

Sincerely yours,

Prof. Dr. Kefu Yu

Prof. Dr. Yuehuan Zhang

Dr. Sanqiang Gong

REVIEWERS' COMMENTS:

Reviewer #2 (Remarks to the Author):

Brief summary: This article investigates an interesting topic at the nexus of coral bleaching, global warming and holobiont chronobiology. The main contribution of this paper is to track the metabolic and rhythmic transcriptional changes while the coral host is bleaching. They managed to distinguish coral transcriptome from the symbiont to identify rhythmic genes in their different experimental conditions (LD, DD, HLD). They identified core pacemaker genes in the host but not in the symbiont. Elevated temperatures disrupted the rhythmic gene (downstream of the pacemaker aka CCG) expression patterns, particularly in the coral host, and led to significant physiological and biochemical changes associated with bleaching. The findings highlight the critical role of diel rhythms in maintaining coral-symbiont symbiosis and suggest that disruption of these rhythms by elevated temperatures contributes to coral bleaching. This study provides valuable insights into the molecular mechanisms underlying coral bleaching and could inform strategies to mitigate the impacts of global warming on coral reefs.

Overall impression of the work: Following the reviewers' recommendations, the authors have made significant efforts to improve the clarity of the manuscript, narrative, and figures. I recommend accepting this article for publication in *Communications Biology*. However, I suggest that the editors check for some spelling mistakes that I noticed in the manuscript. With that, my job as a scientific reviewer is done. Good work!

Raphael Aguillon

Corrections to the manuscript

Dear Reviewers,

We appreciate the time and efforts by the reviewers in reviewing the manuscript entitled **“Diel transcriptional responses of coral-Symbiodiniaceae holobiont to elevated temperature ”**(Manuscript ID: COMMSBIO-23-4034A). The newly submitted manuscript has made point-to-point revision according to the reviewer’s comments. In the revised manuscript, all corrections have been marked in red. The detailed responses and corrections are listed below.

Reviewer #2 (Remarks to the Author):

Brief summary: This article investigates an interesting topic at the nexus of coral bleaching, global warming and holobiont chronobiology. The main contribution of this paper is to track the metabolic and rhythmic transcriptional changes while the coral host is bleaching. They managed to distinguish coral transcriptome from the symbiont to identify rhythmic genes in their different experimental conditions (LD, DD, HLD). They identified core pacemaker genes in the host but not in the symbiont. Elevated temperatures disrupted the rhythmic gene (downstream of the pacemaker aka CCG) expression patterns, particularly in the coral host, and led to significant physiological and biochemical changes associated with bleaching. The findings highlight the critical role of diel rhythms in maintaining coral-symbiont symbiosis and suggest that disruption of these rhythms by elevated temperatures contributes to coral bleaching.

This study provides valuable insights into the molecular mechanisms underlying coral bleaching and could inform strategies to mitigate the impacts of global warming on coral reefs.

Overall impression of the work: Following the reviewers' recommendations, the authors have made significant efforts to improve the clarity of the manuscript, narrative, and figures. I recommend accepting this article for publication in Communications Biology. However, I suggest that the editors check for some spelling mistakes that I noticed in the manuscript. With that, my job as a scientific reviewer is done. Good work!

Response: Thanks for the meticulous checking and helpful suggestions. The spelling and grammar mistakes have been checked throughout the manuscript again, and all mistakes have been revised in newly submitted manuscript.

1. line 81: “Physiology and metabolism rhythmic of coral holobiont” has been revised into “Physiology and metabolism rhythms of coral-Symbiodiniaceae holobiont”.
2. line 87 : “rhythmic oscillations (24-h)” to “rhythmic oscillations (24-hour)”.
3. line 113: “as cadidate clock” to “as candidate clock”
4. line 128: “cadidate clock” to “candidate clock”.
5. line 161: “cadidate clock” to “candidate clock”.
6. line 214-215: “However, we observed that elevated temperature disrupted this diel rhythmicity of gene transcription observing” to “However, we noted that elevated temperature disrupted this diel rhythmicity of gene transcription observed”.
7. line 251: “a acclimatization” to “an acclimatization”.
8. line 258: “light: dark” to “light:dark”.

9. line 470: “Physiology and metabolism rhythmic” to “Physiology and metabolism rhythms”.

10. line 490: “Overall oscillation of rhythmic genes in” to “Overall oscillation of rhythmic genes in the”.

11. line 506: “Diel oscillation of candidate clock genes in ” to “Diel oscillation of candidate clock genes in the”.

12. line 525: “were illustrated” to “was illustrated”.

13. Table 1: “Consinor” to “Cosinor”.

Looking forward to hearing from you and with best wishes.

Sincerely yours,

Prof. Dr. Kefu Yu

Prof. Dr. Yuehuan Zhang

Dr. Sanqiang Gong